# Integrating Continuous Transepithelial Flux Measurements into an Ussing Chamber Set-Up

**DOI:** 10.3390/ijms25042252

**Published:** 2024-02-13

**Authors:** Çlirim Alija, Lukas Knobe, Ioanna Pouyiourou, Mikio Furuse, Rita Rosenthal, Dorothee Günzel

**Affiliations:** 1Clinical Physiology/Nutritional Medicine, Medical Department, Division of Gastroenterology, Infectiology, Rheumatology, Charité–Universitätsmedizin Berlin, 12203 Berlin, Germany; clirim.alija@charite.de (Ç.A.); lukas.knobe@charite.de (L.K.); ioanna.pouyiourou@charite.de (I.P.); rita.rosenthal@charite.de (R.R.); 2Division of Cell Structure, National Institute for Physiological Sciences, Okazaki, Aichi 444-8787, Japan; furuse@nips.ac.jp

**Keywords:** transepithelial flux, paracellular fluorescent marker, Ussing chamber, automatized flux measurement, tight junction, osmotic stress, claudin

## Abstract

Fluorescently labelled compounds are often employed to study the paracellular properties of epithelia. For flux measurements, these compounds are added to the donor compartment and samples collected from the acceptor compartment at regular intervals. However, this method fails to detect rapid changes in permeability. For continuous transepithelial flux measurements in an Ussing chamber setting, a device was developed, consisting of a flow-through chamber with an attached LED, optical filter, and photodiode, all encased in a light-impermeable container. The photodiode output was amplified and recorded. Calibration with defined fluorescein concentration (range of 1 nM to 150 nM) resulted in a linear output. As proof of principle, flux measurements were performed on various cell lines. The results confirmed a linear dependence of the flux on the fluorescein concentration in the donor compartment. Flux depended on paracellular barrier function (expression of specific tight junction proteins, and EGTA application to induce barrier loss), whereas activation of transcellular chloride secretion had no effect on fluorescein flux. Manipulation of the lateral space by osmotic changes in the perfusion solution also affected transepithelial fluorescein flux. In summary, this device allows a continuous recording of transepithelial flux of fluorescent compounds in parallel with the electrical parameters recorded by the Ussing chamber.

## 1. Introduction

In studies investigating epithelial transport properties, flux measurements of fluorescence dyes, such as sodium fluorescein (uranine), or fluorescently labelled compounds, such as FITC-dextrans of various sizes, are frequently employed to assess paracellular permeability for these substances and thus the size selectivity of the paracellular pathway [1,2,3,4]. If combined with the measurement of the transepithelial resistance (TER), flux measurements can be utilized to quantify the partial resistances of the paracellular and transcellular transport pathway [5].

Depending on the experimental set-up, however, flux measurements can be affected by various confounding factors. If filter inserts are incubated in well plates, unstirred layer effects due to lacking or inefficient perfusion may restrict transepithelial diffusion of the marker molecule. Markers, such as sodium fluorescein, dissociate in aqueous solutions. The pKa values of the fluorescein anion have been estimated to be 4.36 and 6.38 for pKa_1_ and pKa_2_, respectively, indicating that at physiological pH values, the predominant form is that of a dianion [6]. As a consequence, fluorescein flux will be affected by changes in transepithelial voltage. Thus, the activation of electrogenic transepithelial transport processes that affect transepithelial voltage will lead to alterations in fluorescein flux that may be falsely interpreted as alterations in the paracellular pathway, if the transepithelial voltage is not clamped throughout the experiment.

These problems can be circumvented if flux measurements are carried out in classical Ussing chambers. Ussing chambers consist of two hemi-chambers that are separated by an epithelial cell layer or tissue. Originally, these chambers were designed to remove all transepithelial gradients (pressure, voltage, concentration) that might drive passive, paracellular transport and thus to isolate transepithelial net charge transport by recording the current (short-circuit current, Isc) needed to clamp the transepithelial voltage to 0 mV [7]. The Ussing chamber can, however, also be utilized to investigate properties of the paracellular pathway, e.g., measuring the transepithelial flux of a paracellular marker substance, i.e., a substance that is not actively transported through the cells [8]. This substance, e.g., the above-mentioned fluorescence dye, is added to one of the hemi-chambers (donor compartment), and samples are collected from the other hemi-chamber (acceptor compartment) at regular intervals. Fluorescence intensity is analyzed, e.g., by means of a plate reader in order to calculate paracellular flux and, from this, paracellular permeability for the analyzed substance.

The advantage of utilizing Ussing chambers is that the two hemi-chambers are constantly perfused, e.g., through a bubble-lift system [9,10,11], and that experiments can be carried out under permanent voltage clamp conditions [5,12,13,14,15]. However, using such systems, other difficulties will be encountered. As samples are drawn from the acceptor side at regular intervals, this will either result in a step-by-step volume decrease at the acceptor side or in a step-wise dilution of the marker substance concentration on the acceptor side, if the sample volume is replaced by fresh solution. In both cases, alterations on the acceptor side have to be taken into consideration when finally calculating fluxes and permeabilities [5,16]. Furthermore, if samples are drawn at short intervals, time resolution will be high, but even a short delay during sampling will result in a sizeable error (e.g., when sampling every 5 min, a 30 s delay will already result in a 10% error). On the other hand, sampling at longer time intervals causes a loss of time resolution, and rapid alterations may not be detected.

To address these difficulties in flux measurements, it was the aim of the present study to establish a device for continuous, automatized flux measurements and integrate it into an Ussing chamber set-up. As a proof of concept, the device was tested on cell layers with greatly differing para- and transcellular transport properties: the colonic cell line HT-29/B6, in which transcellular Cl^−^ secretion can be activated by the application of forskolin [5,17], and MDCK quintuple knockout cells (claudin quin KO), MDCK II cells in which the five most abundant claudins were knocked out [3] and stably transfected to overexpress different claudins, resulting in high (claudin-4) or low (claudin-2, claudin-8) transepithelial resistance with (claudin-2) or without (claudin-8) strong paracellular charge selectivity.

## 2. Results

### 2.1. Calibration of Fluorescein-Induced Output Signal

For calibration of our device, the two hemi-chambers of an Ussing chamber were separated with parafilm. One shank of the Ussing chamber was filled with 10 mL of standard Ringer’s solution. At regular time intervals, defined volumes of a 10 µM solution of Na_2_-fluorescein dissolved in standard Ringer’s solution were added, and the converted output signal of the photosensor (in V) was recorded on a flatbed chart recorder. Directly prior to the addition of the next dose of fluorescein solution, the previously added volume was removed, to adjust the total volume within the chamber again to 10 mL.

Figure 1a shows an original recording of a calibration procedure. In Figure 1b, the output signal (in V) is plotted against the fluorescein concentration. The results show that an increment of 1 nM can be reliably measured and that the output signal of our device is linear up to a concentration of at least 150 nM. This linear dependence confirms the validity of this new device for transepithelial flux measurement. For comparison, a calibration curve for a commercial plate reader is shown.

### 2.2. Linear Dependence of Fluorescein Flux on the Donor Concentration

Permeability of a cell layer for a certain substance is defined as the ratio of the flux of this substance divided by its concentration gradient (Δc) across the cell layer [18]. In the current setting, this implies that the fluorescein flux across a cell layer should depend linearly on the fluorescein concentration within the donor compartment. To test this assumption, fluorescein flux was determined in HT-29/B6 cell layers treated with EGTA to increase permeability. As shown in Figure 2, a step-wise doubling and tripling of the initial fluorescein donor concentration of 100 µM (as previously employed by [5,19]) resulted in a doubling and tripling of the recorded fluorescein flux.

### 2.3. Fluorescein Flux Is Affected by Manipulation of the Paracellular Pathway

To test the applicability of the device, experiments were carried out on MDCK quintuple knockout cells (claudin quin KO; for description see Section 4.1) stably transfected to overexpress the tight junction proteins claudin (Cldn)4 or Cldn8. Non-transfected claudin quin KO cells do not form tight junction strands, as they lack the five most abundant endogenous members of the claudin protein family [3]. Figure 3 shows the original recording of an experiment on claudin quin KO+Cldn4 cells (a) and claudin quin KO+Cldn8 cells (b). Whereas Cldn4 sealed the paracellular pathway against fluorescein permeation, Cldn8-overexpressing cell layers exhibited a very high fluorescein permeability. After assessing basal permeability, the paracellular pathway was manipulated by the application of 100 µL 130 mM EGTA solution to the apical and basolateral side. Chelation of extracellular Ca^2+^ with EGTA caused an immediate loss of barrier function, as indicated by the increase in fluorescein flux in both Cldn4- and Cldn8-overexpressing cell layers (Figure 3c).

In the initial test experiments with our device, shown in Figure 3, parallel TER measurements had not yet been established. However, confluence of the cell layers on the filter supports were confirmed by TER measurements with chopstick electrodes prior to mounting in the Ussing chamber (claudin quin KO+Cldn4 cell layer 615 Ω∙cm^2^; claudin quin KO+Cldn8 cell layer 36 Ω∙cm^2^). Conventional flux measurements with discrete samples analyzed in a plate reader yielded similar flux values for claudin quin KO+Cldn4-overexpressing cell layers and somewhat higher flux values for claudin quin KO+Cldn8-overexpressing cell layers (see Appendix A).

### 2.4. Fluorescein Flux Is Not Affected by Manipulation of the Transcellular Pathway

Fluorescein is commonly used as a paracellular marker but has been reported to be actively transported by organic anion transporters in some cell lines [20]. To test whether alterations in transcellular transport affect fluorescein flux in HT-29/B6 cells, Cl^−^ secretion was induced in these cells by the application of forskolin to the apical side of the cell layer (final concentration of 10 µM, as previously used, e.g., by [5,17]). This caused a distinct reduction in TER (by 187 ± 27 Ω∙cm^2^; *n* = 3) and increase in Isc (by 22.1 ± 6.8 µA/cm^2^; *n* = 3) which was, however, not accompanied by an alteration of fluorescein flux (Figure 4). Subsequent alteration of the paracellular pathway by an application of EGTA caused a further reduction in TER (by 511 ± 29 Ω∙cm^2^; *n* = 3) together with a distinct increase in fluorescein flux.

### 2.5. Osmotic Effects on Paracellular Fluorescein Flux

The paracellular pathway is not only governed by tight junctions but also by the properties of the lateral space. Osmotic cell swelling or shrinkage may alter the accessibility of the lateral space and thus affect paracellular transport. To test this hypothesis, mannitol at a final concentration of 500 mM was added to either the apical or the basolateral side of the epithelium. Figure 5a shows that the addition of mannitol to the basolateral side of claudin quin KO cell layers stably transfected to overexpress Cldn2 led to the expected reduction in TER and increase in fluorescein flux. Unexpectedly, however, application of mannitol to the apical side caused an increase in TER and a decrease in fluorescein flux (Figure 5b).

## 3. Discussion

The present study serves as a proof of principle for continuous flux measurements of fluorescence dyes or fluorescently labelled substances by a device integrated in an Ussing chamber. The device thus allows time-resolved, intervention-free flux measurements in parallel to the electrical parameters (TER, transepithelial potential, short-circuit current) recorded by the Ussing chamber. Sensitivity and detection limit are comparable to a plate reader when measuring duplicates of ~150 µL samples in a 96-well plate, as previously described, e.g., by [5,16,21,22].

Fluorescently labelled compounds are often used to determine the size selectivity of the paracellular pathway of different epithelia. The comparability of flux measurements from different studies, however, is often hampered by the different donor concentrations used in these studies. If flux is truly paracellular and thus passive, flux values can be easily normalized by dividing the flux by the applied concentration gradient, Δc, of the marker compound. The resulting value is the paracellular permeability for this compound, usually specified with the unit 10^−6^ cm/s (permeability = flux [µmol/h/3.6/cm^2^] / Δc [mmol/l = µmol/cm^3^]) [8,18] and can be used to directly compare values from different studies. If, however, part of the transport is transcellular, as, e.g., demonstrated for fluorescein flux [6] and for 4 kDa FITC-dextran flux [23] in Caco-2 cell layers, then this assumed proportionality between Δc and flux would no longer hold true, as an active transport component would be saturable. In our proof-of-concept experiments shown in Figure 2, to avoid potential confounding effects, we increased paracellular permeability by the application of EGTA and were able to demonstrate that, when flux is truly paracellular, it indeed depends linearly on Δc.

Alterations in TER are often taken as an indication for alterations of the paracellular barrier (see, e.g., [24]). However, TER may be equally affected by stimulation of transcellular transport (opening of ion channels in the plasma membrane). Whereas a decrease in TER due to a paracellular defect should be accompanied by an increase in fluorescein flux, this should not be the case for an activation of plasma membrane ion channels. For demonstration, we used the long-known stimulation of chloride secretion by forskolin application in HT-29/B6 cell layers [25]. The results demonstrate that parallel, real-time TER and flux measurements are a powerful tool to directly distinguish between transcellular and paracellular effects during an experiment.

The physiological role of Cldn4 and Cldn8 has been a matter of debate for many years [26,27,28,29,30,31]. Increases in TER have been reported upon upregulation of these claudins in various cell systems (Cldn4 [30,32,33], Cldn8 [31,34,35]). However, human Cldn4 and Cldn8 have been found to be unable to form tight junction strands when overexpressed in Cos-7 cells [36]. Others studies concluded that Cldn4 might act as a paracellular channel, either on its own [28,37] or in combination with Cldn8 [29]. In contrast, barrier-forming properties of dog Cldn4 were demonstrated by Furuse et al. [38], who reported TER values of up to 4000 Ω∙cm^2^ when overexpressing dog Cldn4 in claudin quin KO cells. Average TER values of about 2500 Ω∙cm^2^ were obtained in our experiments with claudin quin KO cells expressing mouse claudin-4, whereas the TER was very low in claudin quin KO cells expressing mouse claudin-8 (averaging about 20 Ω∙cm^2^; for baseline values, see Appendix A; for an example, see Appendix A). Our present results demonstrate clearly that mouse Cldn4 is also able to form a paracellular barrier in claudin quin KO cells, as it greatly restricts transepithelial fluorescein flux. Many factors (source species, expression system, transfection efficiency and thus amount of claudin synthesized by the cells, presence or absence of a tag) may contribute to the conflicting results and necessitate further investigations.

Overall, fluxes determined for Cldn4-overexpressing cell layers compared well to fluxes obtained with discrete samples evaluated in a plate reader (Appendix A). In contrast, fluxes measured in Cldn8-overexpressing cell layers were lower than those obtained with a plate reader (Appendix A). This may be due to the longer duration of flux experiments necessary when collecting discrete samples (Appendix A). The increase in fluorescein flux upon Ca^2+^ chelation may suggest that the Cldn8-based barrier is not negligible. However, it cannot be ruled out that the observed effect was due to other tight junction proteins, such as previously demonstrated for the junctional adhesion protein JAM-A, which was demonstrated to restrict macromolecular paracellular passage [3]. This will be the subject of a future study.

Cldn2 has long been known to act as a paracellular channel for cations [39] and water [14] and is, e.g., involved in Na^+^ and water reabsorption in the proximal tubule of the kidney [40,41]. We originally included cells overexpressing Cldn2; due to the high paracellular cation permeability, they display a low TER but also a low permeability to anions (such as fluorescein) and uncharged molecules (e.g., mannitol; [39]).

In a final set of test experiments, we followed up an observation that induction of an apical to basolateral osmotic gradient across a Cldn2-overexpressing claudin quin KO cell layer caused an increase in TER, whereas a gradient in the opposite direction resulted in a decrease in TER. Similar observations were published in the 1970s and 1980s [42,43,44] and attributed to an opening or collapsing of the lateral space. This induced us to repeat our experiments in a setting that allowed a combined fluorescein flux and TER measurement. The results (Figure 5) confirmed our hypothesis that the changes in TER would be accompanied by alterations in the paracellular pathway. The results stress the importance of the lateral space for paracellular transport, which is by no means only dependent on the tight junction properties.

In future, our prototype device will have to be further developed. Most importantly, the output will have to be digitized and directly recorded, together with the electrical parameters. With this done, we envision that the device will be a powerful tool to be used not only on cell culture but also on tissue samples, to examine mechanisms of paracellular barrier alterations in physiology and in disease.

## 4. Materials and Methods

### 4.1. Cell Culture

MDCK II cells were a gift from Masayuki Murata (Tokyo Institute of Technology, Tokyo, Japan) to the Furuse lab. From these cells, MDCK II quintuple knockout cells (claudin quin KO) had been previously generated by sequential knockout of the five major claudins (Cldn1, -2, -3, -4, -7 [3]). These quintuple knockout cells (claudin quin KO), i.e., MDCK II cells, from which the five major claudins (Cldn1, -2, -3, -4, -7) had been knocked out [3], were stably transfected to overexpress mouse Cldn2 and characterized as described previously [38]. To establish claudin quin KO cells stably overexpressing Cldn4 or Cldn8, cDNAs of mouse *Cldn4* or *Cldn8* [45], subcloned into a mammalian expression vector, pCAGGS containing a neomycin-resistance gene [46], were tranfected into claudin quin KO cells. Cell colonies resistant to 300 µg/mL of G418 were picked, and the expression of each claudin was confirmed by immunofluorescence staining with anti-Cldn4 or anti-Cldn8 antibodies [45].

Transfected claudin quin KO cells were cultured in DMEM(1x) + GlutaMAX-I medium supplemented with 10% fetal calf serum (Thermo Fisher Scientific, Waltham, MA, USA), 1% penicillin/streptomycin (Sigma-Aldrich, Taufkirchen, Germany), and 300 µg/mL G418 (Biochrom, Berlin, Germany) at 37 °C, 5% CO_2_ in a humidified atmosphere. The medium was replaced every two to three days. Cells were seeded onto PCF 0.4 µm pore size filter supports (Millipore, Schwalbach, Germany) at a density of 150,000–200,000 cells per filter (about 10^6^ cells/mL). Confluent, fully differentiated cell layers were thus obtained and used after 7 days (claudin quin KO cells expressing Cldn2, passage 2 to 6; claudin quin KO cells expressing mouse Cldn4 or Cldn8, passage 2 to 9).

The colonic cell line HT-29 was originally obtained from ATCC, and the mucus secreting subclone HT-29/B6 was generated at the Institute of Clinical Physiology (Berlin, Germany), as previously described [47,48]. HT-29/B6 cells were cultured in RPMI 1640 medium (Thermo Fisher Scientific) supplemented with 10% fetal calf serum (Thermo Fisher Scientific) and 1% penicillin/streptomycin (Sigma-Aldrich) at 37 °C, 5% CO_2_ in a humidified atmosphere. The medium was replaced every two to three days. Cells were seeded onto PCF 3.0 µm pore size filter supports (Millipore) at a density of 7 × 10^5^ cells/cm^2^. Confluent, fully differentiated cell layers (passage 10–14) were thus obtained and used after 7–9 days.

Before the filter supports were mounted in Ussing chambers, confluence of the cell layers was routinely tested by measuring TER with chopstick electrodes (STX2 electrodes and EVOM amplifier, WPI, Friedberg, Germany).

### 4.2. Solutions

The standard Ringer’s solution contained the following (in mM): NaCl, 113.6; Na_2_HPO_4_, 2.4; NaH_2_PO_4_, 0.6; KCl, 5.4; NaHCO_3_, 21; MgCl_2_, 1.2; CaCl_2_, 1.2; Glucose, 10. The solution was constantly equilibrated with carbogen gas (5% CO_2_ in 95% O_2_) to obtain a constant pH value of 7.4. A solution of 100 mM sodium fluorescein (37.6 mg/mL; Sigma-Aldrich) in distilled water was prepared. This stock solution was diluted 1:10,000 in standard Ringer’s solution to obtain a 10 µM solution for calibration. For comparison, a plate reader (TECAN infinite M200, Männedorf, Switzerland) was used to determine fluorescence signals (96-well plate, Carl Roth, Karlsruhe, Germany, 150 µL solution per well).

The stock solution for forskolin (Sigma-Aldrich) had a concentration of 10 mM in Dimethyl sulfoxide (DMSO, Carl Roth). EGTA was dissolved in distilled water, and the pH was adjusted to 7.4 with NaOH to obtain a 130 mM stock solution.

### 4.3. Ussing Chamber Set-Up

An Ussing chamber set-up (custom-made, medical-technical laboratories, Charite’ – Universitätsmedizin Berlin, Germany), specifically designed for mounting Millicell filter supports as previously described by [25], was employed for electrophysiological measurements. Current electrodes were ring-shaped silver wire electrodes, and voltage electrodes were Ag/AgCl pellets (WPI) connected to agar bridges (glass capillaries filled with 3% agar prepared in 0.5 M KCl), as previously described by [47,48]. Electrodes were connected to a PSM1700 phase sensitive multimeter (Newtons4th Ltd., Leicester, UK). Filter supports were mounted in the Ussing chamber connected to water-jacketed gas lifts, and 10 mL standard Ringer’s solution was added on each side. The temperature was kept at 37 °C, and constant perfusion and pH was ensured by permanent bubbling with carbogen gas. After mounting, cells were allowed to equilibrate for ≥10 min before experiments were started. For continuous fluorescein flux measurement, the Ussing chamber was fitted with the flux measurement device described below.

### 4.4. Flux Measurement Device

The flux measurement device consisted of the following components: Photodiode (BPW 34 photodiode), LED (5 mm LED, blue, 11,000 mcd, 30°), and optical filter (FEL0500, longpass 500 nm filter, Thorlabs GmbH, Bergkirchen, Germany). Flow-through chambers and light-proof shielding were constructed using the Fusion 360 (Autodesk, San Francisco, CA, USA; https://www.autodesk.de/products/fusion-360/overview, accessed on 10th February 2024; last access February 10th, 2024) in combination with PreForm (Formlabs, Berlin, Germany; https://formlabs.com/de/software/preform/; last accessed on 10th February 2024) and a 3D printer using black bio-compatible resin (Formlabs Form 2 3D printer and Formlabs BioMed Black resin). Windows of the flow-through chambers were covered with coverslips glued to the plastic rim with instant adhesive and additionally sealed with nail polish (Figure 6).

The electric circuit is depicted in Figure 7. The circuit is powered by a 24 V power supply. OP07, together with the resistors R2 and R3, generates a voltage of ~4.3V, which powers the LED and also serves as a reference voltage for the transimpedance amplifier. The transimpedance amplifier converts the photocurrent of the photodiode to a voltage output with a factor of 36 MΩ. The resulting output signal is recorded on a L250E flatbed chart-recorder (Van Renesse, Den Haag, The Netherlands).

The fully assembled flux measurement device integrated into the Ussing chamber set-up is shown in Figure 8.

### 4.5. Statistics

All values are given as mean ± SEM. In Figure 3, statistical analysis was performed with an unpaired, two-sided Student’s *t*-test.

## Figures and Tables

**Figure 1 ijms-25-02252-f001:**
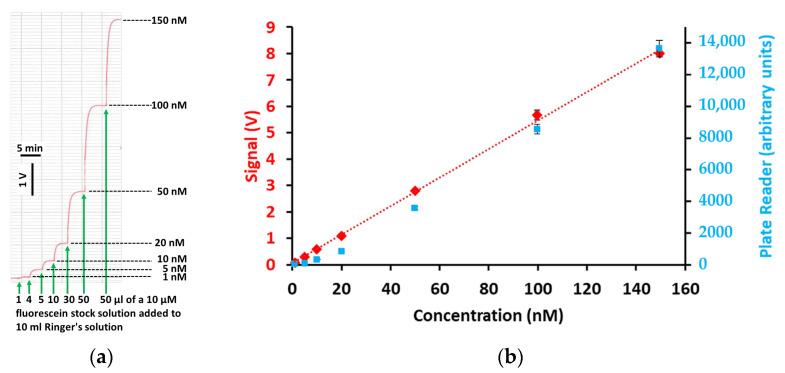
(**a**) Original recording of a calibration experiment. Fluorescein was added cumulatively (green arrows) from a 10 µM stock solution to 10 mL of Ringer’s solution, and the output voltage of our device was recorded on a flatbed chart recorder. Directly prior to each addition of fluorescein, the volume was adjusted to 10 mL by removing the previously added volume. The trace shows that even an increment of 1 nM can be reliably detected. (**b**) Plotting the output voltage of our device (red) and of a plate reader (blue) against the concentration (mean ± SEM, *n* = 4 to 7 for individual data points) demonstrated linearity of the calibration for our device within the tested concentration range of 1 nM and 150 nM (dotted line, linear regression; R^2^ = 0.996), whereas the plate reader signals deviated from linearity at low concentrations.

**Figure 2 ijms-25-02252-f002:**
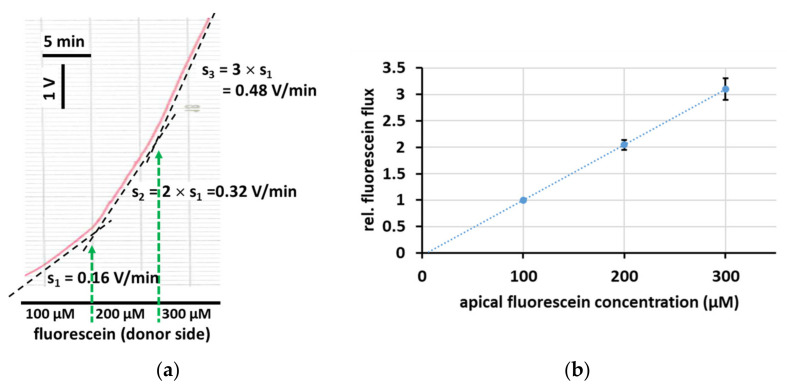
Linear dependence of fluorescein flux on its gradient. (**a**) Apical to basolateral fluorescein flux across a HT-29/B6 cell layer (treated with EGTA to increase permeability) was recorded in the presence of 100 µM fluorescein on the donor side. As indicated by green arrows, fluorescein concentration on the donor side was subsequently increased to 200 and 300 µM. The step-wise increase in the fluorescein gradient caused a proportional step-wise increase in the slope s (dotted lines, V/min), i.e., a linear increase in the fluorescein flux. (**b**) Average of three such experiments (fluorescein flux relative to the flux observed at 100 µM apical fluorescein, plotted against the apical fluorescein concentration; mean ± SEM). Dotted line: linear regression, R^2^ = 0.957.

**Figure 3 ijms-25-02252-f003:**
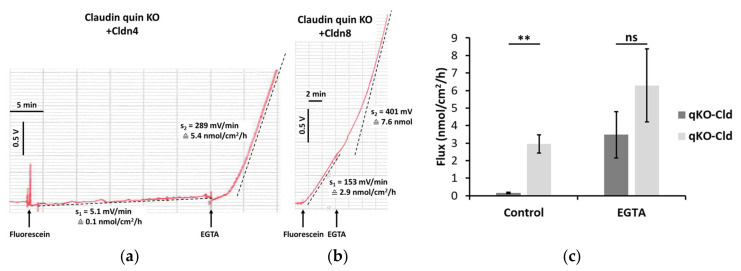
(**a**,**b**) Original recording of fluorescein flux measurement (increase in fluorescein concentration on the acceptor side per time unit) across cell layers of claudin quin KO overexpressing Cldn4 (**a**) and Cldn8 (**b**). Slope s (V/min) of the linear parts of the recordings were determined (dotted line) and converted into fluxes (nmol/cm^2^/h) using the calibration shown in Figure 1. Under control conditions in standard Ringer’s solution, flux was in the order of 0.1 nmol/cm^2^/h in a Cldn4-overexpressing cell layer and about 3 nmol/cm^2^/h in a Cldn8-overexpressing cell layer. In both experiments, flux increased distinctly after the addition of EGTA (final concentration 1.3 mM) to the apical and basolateral side of the cell layer (5.5 and 7.5 nmol/cm^2^/h, respectively). (**c**) Averaging the fluxes of three such experiments shows a highly significant (*p* < 0.01) difference between fluorescein fluxes across Cldn4- and Cldn8-overexpressing claudin quin KO cell layers. After application of EGTA, fluorescein flux increased considerably. Under these conditions, there was no significant difference between Cldn4- and Cldn8-overexpressing claudin quin KO cell layers, indicating a complete loss of barrier function. Statistical analysis was performed with an unpaired, two-sided Student’s *t*-test, ** *p* < 0.01, ns—not significant.

**Figure 4 ijms-25-02252-f004:**
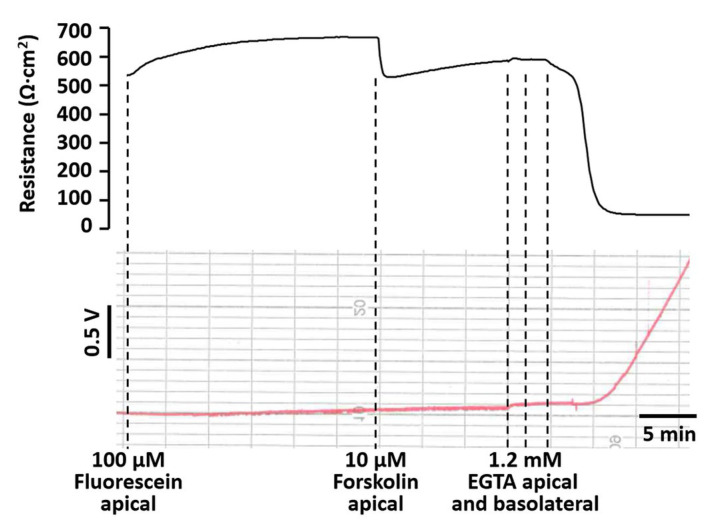
Fluorescein flux is not affected by a reduction in TER caused by the stimulation of a transcellular transport pathway. TER and apical to basolateral fluorescein flux (induced by application of 100 µM fluorescein to the apical compartment of an Ussing chamber) were recorded simultaneously on a HT-29/B6 cell layer before and after induction of transcellular chloride secretion by apical application of forskolin (10 µM final concentration). Forskolin caused a reduction in TER, but fluorescein flux remained unaltered. Subsequently, the paracellular pathway was manipulated by a step-wise addition of EGTA to the apical and basolateral compartments (final concentration of 1.2 mM). Similar to that displayed in Figure 3, this caused a further drop in TER, which was accompanied by an increase in fluorescein flux.

**Figure 5 ijms-25-02252-f005:**
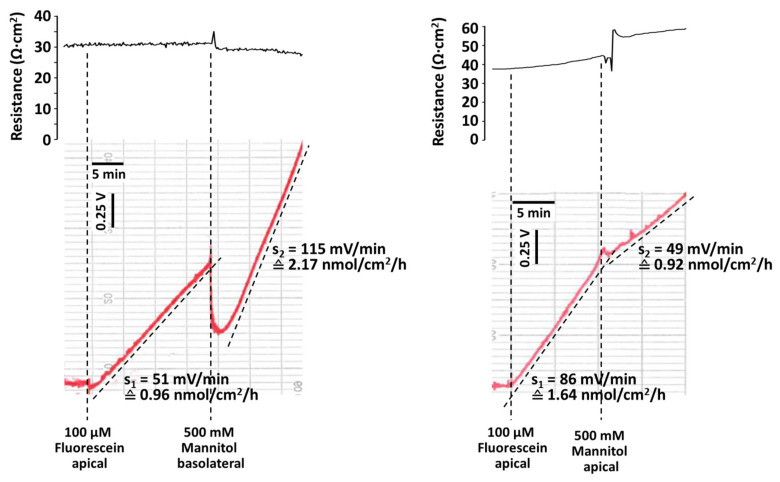
(**a**,**b**) Original recording of TER and apical to basolateral fluorescein flux measurement across cell layers of claudin quin KO stably transfected to overexpress claudin-2. The Ussing chamber was filled with 10 mL on each of the apical and basolateral sides. Where indicated, 10 µL of a 100 mM fluorescein stock solution was added to the apical side, inducing an increase in signal on the basolateral side. (**a**) Under control conditions in standard Ringer’s solution, flux was 0.96 nmol/cm^2^/h. For mannitol application, 5 mL of the basolateral Ringer’s solution was replaced by 5 mL of a stock solution containing 1 M mannitol in Ringer’s solution. This decreased the fluorescein concentration by 50%, causing an instantaneous drop in the recorded signal. Following the application of mannitol, TER dropped, and fluorescein flux increased distinctly to 2.17 nmol/cm^2^/h. (**b**) Under control conditions in standard Ringer’s solution, flux was 1.64 nmol/cm^2^/h. For mannitol application, 5 mL of the apical Ringer’s solution was replaced by 5 mL of a stock solution containing 1 M mannitol and 100 µM fluorescein in Ringer’s solution, ensuring constant fluorescein concentration on the apical side. Following the application of mannitol, TER increased, and fluorescein flux decreased distinctly to 0.92 nmol/cm^2^/h.

**Figure 6 ijms-25-02252-f006:**
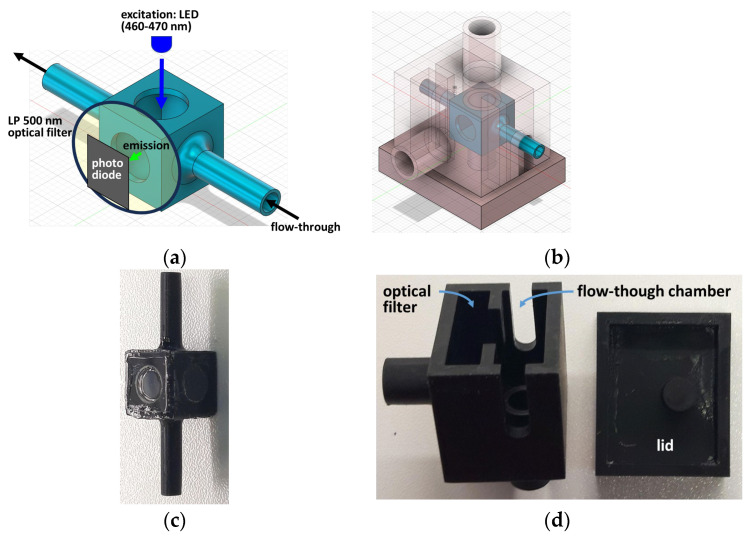
Digital design and 3D printout of components. (**a**) Digital design of flow-through chamber, with position of LED, optical filter, and photodiode indicated. (**b**) Digital design of light-shielding encasement, with position of flow-through chamber indicated. (**c**) 3D printout of flow-through chamber. Windows covered with coverslips. (**d**) 3D printout of light-shielding encasement with compartments for the optical filter and the flow-through chamber and a separate lid.

**Figure 7 ijms-25-02252-f007:**
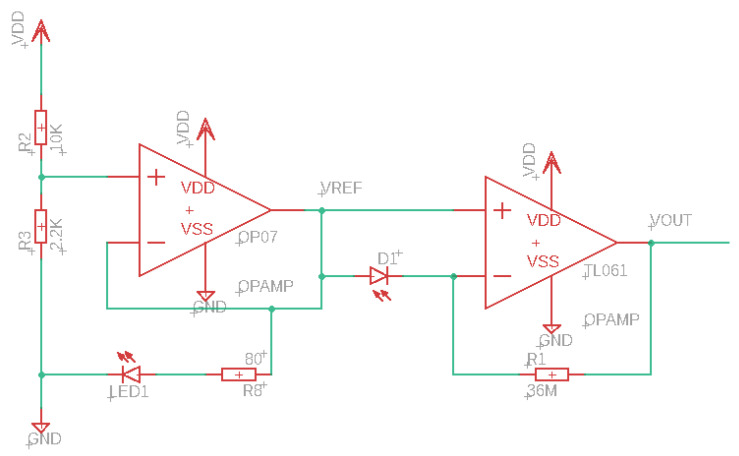
Electric circuit of the unit energizing LED and photodiode, and amplifying photodiode output. The following parts were used: BPW 34 Photodiode, TL061 operational amplifier, OP07 operational amplifier, 12 × 3 MΩ resistors, 10 kΩ resistor, 2.2 kΩ resistor, 80 Ω resistor, 5 mm LED (blue, 11,000 mcd, 30°), and 24 V power supply.

**Figure 8 ijms-25-02252-f008:**
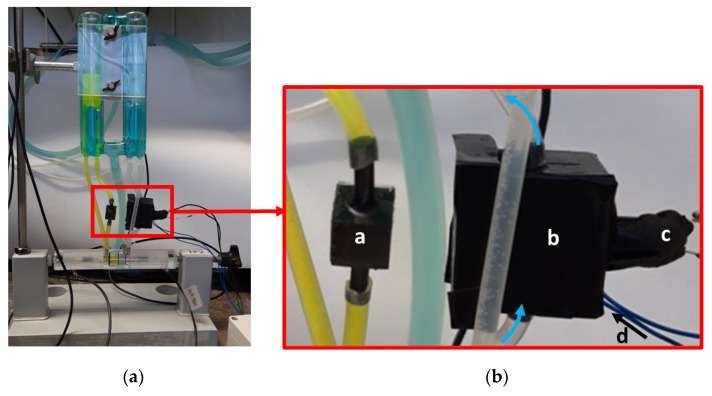
(**a**) Ussing chamber with integrated flux measurement device. (**b**) Detail of (a). Component *a* has the same dimensions as the flow-through chamber used for flux measurements and serves the volume adjustment on the donor side. *b* is the light shielding for the measuring device. It encloses the flow-through chamber (direction of flow indicated by blue arrows), the optical filter, the LED, and the photodiode. *c* is the inlet for the LED, *d* shows the connecting wires for the photodiode, the inlet of which is on the back side.

## Data Availability

Data available on request.

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
