# Peer review of "Integrating Continuous Transepithelial Flux Measurements into an Ussing Chamber Set-Up"

_ijms, 2024, doi:10.3390/ijms25042252_

Round 1

Reviewer 1 Report

Comments and Suggestions for Authors

Integrating continuous transepithelial flux measurements into an Ussing chamber set-up

This is an interesting paper with a novel way of carrying out experiments using Ussing chambers. The paper explains a device which measures changes in fluorescence in the buffer in the Ussing chambers in real time. Traditionally samples are taken at defined time periods and measured on a plate reader. Overall the paper is well written but more explanation is needed in places. The authors need to be aware that not everyone is familiar with the chambers. Authors also need to be aware that this paper is being published in a journal whose readers may not be familiar with how this type of device works and its readouts. 

Introduction

Need to explain what an Ussing chamber is and what types of studies can be carried out using this device. Time as a factor is dependent on the study. Is it drug fluxes or is it permeation enhancers? Other fluorescent markers. P-glycoprotein studies with rhodamine 123. 

Line 52-53: result in a step-by-step volume decrease at the acceptor side, or in a step-wise dilution of  the marker substance concentration on the acceptor side, if the sample volume is replaced 53 by fresh solution. Replacement is the norm and this is factored into any calculation of apparent permeability (Papp). Also need to think about maintaining sink conditions.

Method:

Cell culture: Passage number of cells used and where were they obtained from.

Why did you use these particular cells ? 

What was the baseline resistance or TEER of each cell line? 

aqua dest. - distilled water

Reference for concentration of forskolin and sodium fluorescein used. 

Give sodium fluorescein in mg/ml as well. 

EGTA was used which is a calcium chelator but there was calcium in the Ringer’s solution. Please comment on this. 

Was TEER measured ? You only give results in some of the studies? How did you know if the cells on the filter were ok to use? That the barrier was intact. 

Were electrodes used? If no. could electrodes interfere with the measurement ?

No mention of clapping ?

Were the cells equilibrated after mounting before anything was added ?

Need to state Ussing chamber used and the manufacturer. More detail of the set up. 

No section on how statistics were performed.

Results:

Need to explain better what the read-out of the device is ? Volts ?

Figure 1a: need to describe better what this figure represents. Or maybe regraph. 

At higher concentration of the sodium fluorescein did the authors need to dilute samples in order to read on a plate reader ? Or could this be an issue if people were using higher concentrations of sodium fluorescein. The apical concentration of something like FD4 can be as high as 2.5mg/ml.

Did you have r squared values for figure 2b. 

Did you think about calculating apparent permeability? It is a commonly used parameter. 

Did you look at mass balance ? 

Need to explain figure 2a better. 

Line 104: do you mean section not chapter?

Duration of experiments ?

Need to explain what the different claudins are responsible for. 

Figure 3: is this measurement of the acceptor side?

Figure 3c: You see no difference in the EGTA groups due to variability. n=3 might have been too low. What statistics were performed ? t-test?

If you are looking at forskolin it is short circuit current that will temporarily be affected? Did you measure this? Need to show that increase.

Figure 5a: If you removed half the volume of buffer (5ml) did you do this on both sides at the same time ? If not, this can cause a pressure difference and affect the cells ? It is better practice to make a higher concentration stock and only remove 400 or ul and replace with the marker.

Future studies - tissue ??

Overall more detail and a better explanation of what is happening.

Author Response

Please see separate pdf

Reviewer 2 Report

Comments and Suggestions for Authors

This submission described a device to investigate the continuous transepithelial flux measurements in an Ussing chamber. The experiments were unfolded straight forward and the discussion was meaningful. I recommend a major revision on the following points:

1. In my opinion, the topic was close to the signal processing. The signals before and after processing, which status was closer to the real situation? Whether the main purpose was improving the accuracy or the convenience of application?

2. If the basic pattern of the curve was an “S” shape one, have the authors considered to address the whole curve in several sections and describe the characteristics of each section?

3. For the optical signal treatment, I recommend the authors to cite several biological strategies with protein mechanical approaches (Crit. Rev. Anal. Chem., 2022, 52: 72-92; Anal. Chem., 2022, 94: 4594-4601; Coordin. Chem. Rev., 2021, 445: 214068).

4. The format standardization and language use should be improved. For example, the standard units, some confusing expression.

Comments on the Quality of English Language

Moderate editing needed.

Author Response

This submission described a device to investigate the continuous transepithelial flux measurements in an Ussing chamber. The experiments were unfolded straight forward and the discussion was meaningful. I recommend a major revision on the following points:

Thank you very much for your review!

  1. In my opinion, the topic was close to the signal processing. The signals before and after processing, which status was closer to the real situation? Whether the main purpose was improving the accuracy or the convenience of application?

I fear, this is a misconception, our device simply measures bulk fluorescence in the bath solution of the Ussing chamber. Currently there is no signal processing, the analog signal is recorded on a chart recorder. However, digitization and recording on a computer will be the next step.

  1. If the basic pattern of the curve was an “S” shape one, have the authors considered to address the whole curve in several sections and describe the characteristics of each section?

I am not sure, which curve this statement is referring to? To the data obtained for comparison by using a plate reader (Fig. 1b, blue symbols)? Yes, indeed the characteristics of each section could be evaluated. However, our new device renders this unnecessary, with the additional advantage of a much better time resolution of the flux measurement (continuous measurement as compared to discrete measurements 5 to 10 min apart).

  1. For the optical signal treatment, I recommend the authors to cite several biological strategies with protein mechanical approaches (Crit. Rev. Anal. Chem., 2022, 52: 72-92; Anal. Chem., 2022, 94: 4594-4601; Coordin. Chem. Rev., 2021, 445: 214068).

We do not see, how these publications relate to our work and have therefore chosen not to include them in our paper.

  1. The format standardization and language use should be improved. For example, the standard units, some confusing expression.

We have attempted to standardize units as well as improve language, and do hope that the manuscript is now more consistent.

Reviewer 3 Report

Comments and Suggestions for Authors

This paper is nicely written and the methods are implemented nicely. 

However, I have few comments before recommending for publication in this Journal. 

1. Fluorescently doesn't sound the term used frequently.  Authors can use Fluorescent-tagged etc.

2. What does the author mean by acceptor and donor compartments?

3. What is just calibration in the first result of the paper. It should have couple of words, reflecting the following results precisely.  Like calibration of fluoriscein concentration for ...

4. What kind of cell lines the author have used?

5. What are the feasibility and technical challenges for this experiment in in vivo applications. 

Author Response

This paper is nicely written and the methods are implemented nicely. 

However, I have few comments before recommending for publication in this Journal. 

Thank you very much for your review.

  1. Fluorescently doesn't sound the term used frequently.  Authors can use Fluorescent-tagged etc.

We believe that "fluorescently" is correct (google: > 9 million hits, see e.g. https://www.pepscan.com/custom-peptide-synthesis/peptide-modifications/fluorescently-labeled-peptides/; https://pubmed.ncbi.nlm.nih.gov/32154932/; https://pubmed.ncbi.nlm.nih.gov/6987537/; https://www.sciencedirect.com/science/article/abs/pii/S2452074817301283 )

  1. What does the author mean by acceptor and donor compartments?

In flux experiments, the donor compartment is the compartment (here: the hemichamber of the Ussing chamber) to which the tracer is added. The acceptor compartment is the compartment (hemichamber of the Ussing chamber) into which the tracer diffuses/is transported, i.e. from which samples are collected to determine the transport rate. We are now explaining this in the Introduction.

  1. What is just calibration in the first result of the paper. It should have couple of words, reflecting the following results precisely.  Like calibration of fluoriscein concentration for ...

We have now changed the heading to “Calibration of fluorescein-induced output signal “added more information to the legend of Figure 1a.

  1. What kind of cell lines the author have used?

The cell lines are described in the Methods section, which, unfortunately, is located only at the end of the manuscript. We have now also added more information on the cell lines we employed at the end of the introduction.

  1. What are the feasibility and technical challenges for this experiment in in vivo applications. 

Unfortunately, in vivo applications of Ussing chambers are generally impossible. It is possible, however, to use our set-up on ex vivo tissue (e.g. human intestinal biopsies, mouse or rat intestine, etc.)

Round 2

Reviewer 2 Report

Comments and Suggestions for Authors

The authors have revised the submission rationally. I recommend the acceptance.

Comments on the Quality of English Language

Some minor revision needed.